# QuinNet: Efficiently Incorporating Quintuple Interactions into Geometric Deep Learning Force Fields

**Zun Wang**
Microsoft Research AI4Science
Beijing, China, 100084
`zunwang@microsoft.com`

**Guoqing Liu**
Microsoft Research AI4Science
Beijing, China, 100084

**Yichi Zhou**
Microsoft Research AI4Science
Beijing, China, 100084

**Tong Wang**
Microsoft Research AI4Science
Beijing, China, 100084

**Bin Shao**
Microsoft Research AI4Science
Beijing, China, 100084
`binshao@microsoft.com`

## Abstract

Machine learning force fields (MLFFs) have instigated a groundbreaking shift in molecular dynamics (MD) simulations across a wide range of fields, such as physics, chemistry, biology, and materials science. Incorporating higher order many-body interactions can enhance the expressiveness and accuracy of models. Recent models have achieved this by explicitly including up to four-body interactions. However, five-body interactions, which have relevance in various fields, are still challenging to incorporate efficiently into MLFFs. In this work, we propose the quintuple network (QuinNet), an end-to-end graph neural network that efficiently expresses many-body interactions up to five-body interactions with *ab initio* accuracy. By analyzing the topology of diverse many-body interactions, we design the model architecture to efficiently and explicitly represent these interactions. We evaluate QuinNet on public datasets of small molecules, such as MD17 and its revised version, and show that it is compatible with other state-of-the-art models on these benchmarks. Moreover, QuinNet surpasses many leading models on larger and more complex molecular systems, such as MD22 and Chignolin, without increasing the computational complexity. We also use QuinNet as a force field for molecular dynamics (MD) simulations to demonstrate its accuracy and stability, and conduct an ablation study to elucidate the significance of five-body interactions. We open source our implementation at `https://github.com/Zun-Wang/QuinNet`.

## 1 Introduction

Machine learning force fields (MLFFs) [1, 2, 3, 4, 5, 6] have brought about a transformative shift in molecular dynamics (MD) simulations across diverse fields, including physics, chemistry, biology, and material science. MLFFs enable superior accuracy and reliability by leveraging *ab initio* and experimental data to learn interatomic interactions, while simultaneously keeping the low cost of empirical force fields. As a result, MLFFs provide a powerful tool for accurately modeling molecular

37th Conference on Neural Information Processing Systems (NeurIPS 2023).

interactions, and have the potential to revolutionize our understanding of complex systems at both the atomistic and macroscopic scales.

Recent geometric deep learning potentials, represent atoms in a system as nodes and chemical bonds as edges in a graph while incorporating physical symmetries. These methods have been proven to be data-efficient and effective in modeling many-body interactions [7]. The development track of geometric deep learning potentials is similar to that of empirical force fields, which follow the many-body expansion theory [8, 9, 10]. Previous studies [11, 12, 13, 14, 15, 16, 17, 18, 19, 20] have primarily adopted a hierarchical approach to model truncated energy correction terms, enabling the incorporation of higher-order interactions. As higher-order many-body interactions play a critical role in accurately modeling complex molecular systems, this approach has shown promise in leveraging graph neural networks (GNNs) [21] to construct more expressive and accurate models.

Nowadays, GNN models have incorporated many-body interactions up to four-body interactions explicitly [17, 18, 19, 20] and incorporating higher order many-body interactions is still a challenge. The neglect of higher-order many-body interactions may result in incomplete molecular representations [22] and limited accuracy in certain cases [23, 24]. For example, five-body interactions play an important role in various fields such as coarse-grained protein force field [25], organic molecules [26], crystal vibrations [27, 28], electrostatic interaction potentials [29], and so on.

In this paper, we develop a MLFF based on GNNs, called quintuple network (QuinNet), to address the challenge of incorporating higher order many-body interactions, up to quintuple, into molecular representations without introducing more computational complexity. To achieve this, we analyze the topology of diverse many-body interactions and design QuinNet to explicitly represent these interactions efficiently. Our approach outperforms existing methods in terms of accuracy on several large molecular systems (e.g., MD22 and Chignolin dataset) while remaining comparable on small molecules (e.g., MD17 and revised MD17 dataset). Additionally, computational complexity analysis demonstrates the efficiency of our model. Our work has the potential to advance the field of MLFFs by providing methods to incorporate higher order many-body interactions.

## 2 Related Work

The field of MLFFs has witnessed significant progress since the introduction of the first neural network-based framework by Behler and Parrinello [30]. However, the inherent indistinguishability of atoms has posed a challenge in ordering them, which has limited the success of this approach. To overcome this challenge, GNNs have emerged as a promising solution, leveraging the power of graph formalism to focus on relationships among entities rather than individual node properties.

Currently, two mainstream GNN-based methods have been developed for constructing force fields: group theory-based methods and direction-based methods. Group theory-based methods preserve physical symmetries by imposing trainable weights as the representation of related groups [22, 31]. These equivariant models, including Tensor Field Networks [32], Cormorant [33], SE(3)-Transformers [34], NequIP [7], Equiformer [35], and Allegro [36], have demonstrated high performance across diverse tasks. Furthermore, the MACE [37], which was developed hierarchically using a body order expansion manner, has been proposed as a unifying framework of E(3)-equivariant atom-centered interatomic potentials, extending the ACE [38] framework to include methods built on equivariant message passing neural networks (MPNNs).

On the other hand, direction-based methods, such as SchNet [11, 12], DimeNet (DimeNet++) [13, 14], PaiNN [15], ET [16], GemNet [17], SphereNet [18], ComENet [19], and ViSNet [20], explicitly model interatomic interactions by incorporating more geometric information. This progress has motivated us to investigate the potential performance enhancements of explicitly incorporating higher-order interactions into GNN-based force fields.

## 3 Preliminary

### 3.1 GNN-based force fields

The key idea of graph neural networks boils down to a message-passing framework [39], wherein each node iteratively updates its embedding by aggregating messages from its neighboring nodes.

With $h_v$ and $e_{vw}$ denoting node features and edge features respectively, the message-passing layer is defined as

$$m_v^{t+1} = \sum_{w \in \mathcal{N}(v)} M_t(h_v^t, h_w^t, e_{vw}^t),$$
$$h_v^{t+1} = U_t(h_v^t, m_v^{t+1}),$$

(1)

where $M_t$ and $U_t$ are a message function and an update function, respectively. In the realm of GNN-based force field, physical symmetries such as translation, rotation, and permutation symmetries have been demonstrated to be intrinsic and highly effective for model design.

**Group equivariance**   Let $\mathcal{L} : \mathcal{X} \to \mathcal{Y}$ be a function that maps from input space $\mathcal{X}$ to output space $\mathcal{Y}$. We say that $\mathcal{L}$ is equivariant with respect to a group $G$ if for all group element $g \in G$,

$$\mathcal{L} \circ D^{\mathcal{X}}(g) = D^{\mathcal{Y}}(g) \circ \mathcal{L},$$

(2)

where $D^{\mathcal{X}}$ is the representation of group $G$ in the input vector space $\mathcal{X}$.

**Group equivariant convolutional layer**   In recent years, there has been a growing interest in designing group equivariant neural networks due to the importance of respecting relevant symmetries as an inductive bias. Specifically, the group equivariant convolutional layer is defined as follows:

$$\mathcal{L}_{acm_o}^{(l_o)}(\vec{r}_a, V_{acm_i}^{l_i}) := \sum_{m_f, m_i} C_{(l_f, m_f)(l_i, m_i)}^{(l_o, m_o)} \sum_{b \in S} F_{cm_f}^{(l_f, l_i)}(\vec{r}_{ab}) V_{bcm_i}^{l_i},$$

(3)

where the node features are denoted by $V$, and $\vec{r}_{ab}$ represents the relative position vector between point $a$ and point $b$. The subscripts $c$, $i$, $f$, and $o$ denote channel, input, filter, and output, respectively. Additionally, $l$ and $m$ correspond to the orbital quantum number and magnetic quantum number, respectively. This definition of the group equivariant convolutional layer is an important contribution to the development of neural networks that can effectively handle physical symmetries.

## 3.2   Empirical force fields

In classical molecular dynamics simulations, force fields play a crucial role in describing the potential energy $\mathscr{V}(\mathbf{r}^N)$ of a system, where $\mathbf{r}^N$ represents the positions of $N$ particles. A common functional form for force fields is $\mathscr{V}(\mathbf{r}^N) = E_{\text{bonded}} + E_{\text{non-bonded}}$. The bonded energy term captures the deviation of bonds and angles from their equilibrium values, while the non-bonded interactions include long-range forces, such as electrostatic and van der Waals interactions. To represent the bonded interactions hierarchically, bond stretching, angle bending, bond rotation, and other factors can be taken into account.

**Improvement of torsion potential**   The torsion potential is a crucial aspect of molecular modeling, representing the energy associated with the rotation of dihedral angles between bonded atoms. While many of the torsion terms are comprised of only one term from the cosine series expansion [40], it has been discovered that including more than one term is necessary for certain bonds, such as gauche conformations [41, 42]. This involves the use of higher order cosine series, represented by $\sum_{n=0}^{N} \frac{V_n}{2} [1 + \cos(n\omega - \gamma)]$, where $\omega$ denotes the torsion angle and $V_n$ is the barrier height.

**Improper torsions and out-of-plane bending motions**   Experimental observations have shown that in certain molecular systems, the atom connected to a ring tends to remain in the same plane due to the $\pi$-bonding energy. However, the force field discussed in the previous section predicts an equilibrium structure with the atom out of plane. In order to correct this discrepancy, an additional term or terms need to be incorporated into the force field to ensure that the sp$^2$ carbon and its three bonded atoms remain in the same plane. One simple approach is to include an "out-of-plane"

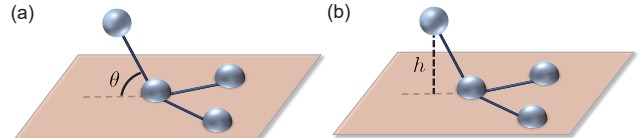

Figure 1: Two methods to represent out-of-plane bending terms: (a) calculating the angle between a bond from the central atom and the plane defined by the central atom and the other two atoms and (b) calculating the height of the central atom above a plane defined by the other three atoms.

bending term [43]. Fig. 1 illustrates two methods that can be used to model the out-of-plane bending contributions. The first method involves calculating the angle between a bond from the central atom and the plane defined by the central atom and the other two atoms, as shown in Fig. 1 (a). The second method calculates the height of the central atom above a plane defined by the other three atoms, as shown in Fig. 1 (b).

### 3.3 Challenge of incorporating higher order many-body interactions into GNNs explicitly

Higher-order many-body interactions could be learned implicitly by stacking more MPNN layers, while incorporating these interactions explicitly into GNNs would improve the performance significantly. Incorporating many-body interactions explicitly into GNN models heavily relies on the selection of appropriate physical quantities, such as bond lengths, bond angles, and dihedral angles. However, for higher order many-body interactions, finding a suitable quantity poses a significant challenge. Moreover, calculating such a physical quantity increases the computational complexity as the order of interactions increases. Therefore, solving this problem while simultaneously reducing computational complexity is of utmost importance.

## 4 Methods

To explicitly incorporate five-body interactions into GNNs, we first analyze the topology of three-, four-, and five-body interactions to identify the key physical quantities required to describe these many-body interactions. Based on this analysis, we design the model architecture to effectively capture and represent these interactions.

### 4.1 Topology of five-body interactions

Five-body interactions have been shown to play a critical role in various cases, such as reproducing specific phenomena in protein systems [25]. While three-body and four-body interactions can be effectively captured by incorporating angular and dihedral angular information, which physical quantity is suitable for representing the five-body interactions explicitly is still a question.

In this work, we propose to use dihedral angles as a unified physical quantity to describe five-body interactions. Fig. 2 illustrates the topology of three-body, four-body, and five-body interactions, respectively. Fig. 2 (a) and (c) show that three-body interactions and improper torsion in four-body interactions could be represented as angles; and Fig. 2 (b) implies that torsion in four-body interactions could be represented as dihedral angles. There are only three different topologies for five-body interactions (Fig. 2 (d)-(f)) and all of them could be represented as dihedral angles. To calculate dihedral angles, we first compute the cosine of the angle between two planes formed by vectors connecting neighboring atoms. For instance, in Fig. 2 (d), atoms $j_1$, $j_2$, $j_3$, and $j_4$ form a neighborhood around atom $i$, and the normal vector of the plane composed of vectors $\vec{r}_{ij_1}$ and $\vec{r}_{ij_2}$ can be computed using their outer product, i.e. $(\vec{r}_{ij_1} \times \vec{r}_{ij_2}) \cdot (\vec{r}_{ij_3} \times \vec{r}_{ij_4})$. The dihedral angle

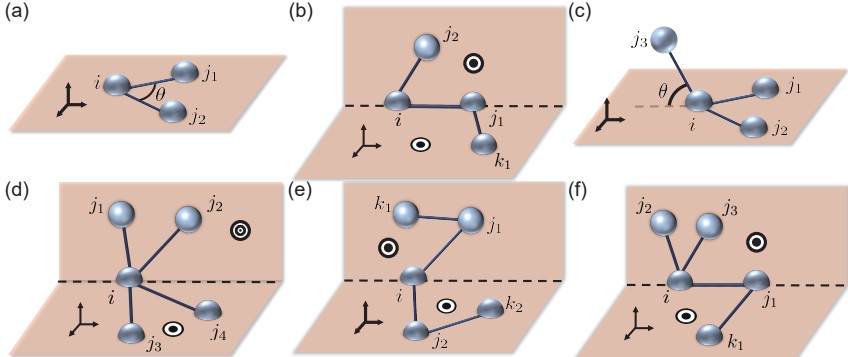

Figure 2: A schematic diagram that describes the topology of (a) three-body interaction (angles), (b) four-body interaction (torsions), (c) four-body interaction (improper torsions), and (d-f) five-body interactions. The marker $\odot$ on a plane represents the normal vector of this plane.

between these two planes can be easily determined as the supplementary angle to the angle between their normal vectors, which can be calculated by taking their inner product. Further details on the relationships between different many-body interactions refer to the Supplementary Materials.

## 4.2 Architecture of QuinNet

Inspired by the topology analysis of many-body interactions, we design the model architecture of QuinNet. The overall schematic diagram of QuinNet is shown in Fig. 3 (a), which explicitly incorporates many-body interactions and reduces computational complexity simultaneously. In particular, we design the QuinNet module, as shown in Fig. 3 (b), where different many-body (i.e., from 3-body to 5-body) interactions are handled separately and linearly combined as the output of the module.

**Three-body interactions**    The three-body interaction, which is related to angular information as depicted in Fig. 2 (a), is handled using the PaiNN [15] approach (refer to Fig. 3 (c)). The equation for the three-body interaction is expressed as:

$$\left\| \sum_{j \in \mathcal{N}_i} \hat{r}_{ij} \right\|^2 = \sum_{j,k \in \mathcal{N}_i} \langle \hat{r}_{ij}, \hat{r}_{ij} \rangle = \sum_{j,k \in \mathcal{N}_i} \cos \alpha_{jik}, \tag{4}$$

where $\hat{r}_{ij}$ denotes the normalized vector of relative position bewteen atom $i$ and $j$, $\vec{r}_{ij}$, and $\langle \cdot, \cdot \rangle$ is inner product. This approach incorporates the representation of bond angular information into node $i$.

**Four-body interactions (torsion)**    The incorporation of torsion interactions in force fields necessitates the use of dihedral angular terms, as depicted in Fig. 2 (b). To address this, we adopt the ViSNet [20] model, which explicitly accounts for torsion energy, as shown in Fig. 3 (b). This is achieved through vector rejection of the direction unit of node $i$, $\vec{v}_i$, which is equivalent to Gram-Schmidt normalization. In 3-dimensional space, the process reduces to calculating the cross product of the normal vector of two planes, as what we summarized in the last section. The torsion angles are then represented via

$$\left( \sum_{j \in \mathcal{N}_i} \hat{r}_{ij} \times \hat{r}_{ij_1} \right) \cdot \left( \sum_{k \in \mathcal{N}_{j_1}} \hat{r}_{jk} \times (-\hat{r}_{ij_1}) \right) = \sum_{\substack{j \in \mathcal{N}_i, \\ k \in \mathcal{N}_{j_1}}} \langle \vec{n}_{ij_1j_2}, \vec{n}_{ij_1k} \rangle, \tag{5}$$

where $\vec{n}_{ijk_1}$ is the normal vector of the plane composed of nodes $i$, $j$, and $k_1$ ($i$, $j$, and $k_1$ are non-colinear). This approach constructs hidden states with dihedral angular information for edge $e_{ij}$.

**Four-body interactions (improper torsion)**    As ViSNet does not incorporate improper torsion term, the QuinNet module incorporates the out-of-plane bending term by computing it in the same manner as Fig. 2 (c), as depicted in Fig. 3 (e). Specifically, the term is computed using the equation

$$\sum_{j \in \mathcal{N}_i} \hat{r}_{ij} \cdot \left[ \left( \sum_{j \in \mathcal{N}_i} \alpha_j \hat{r}_{ij} \right) \times \left( \sum_{j \in \mathcal{N}_i} \beta_j \hat{r}_{ij} \right) \right] = \sum_{j_1,j_2,j_3 \in \mathcal{N}_i} \gamma_{ij_1j_2} \langle \vec{r}_{ij_3}, \vec{n}_{ij_1j_2} \rangle. \tag{6}$$

where the different coefficients $\alpha_j$ and $\beta_j$ are introduced to prevent the term from being zero due to the properties of the cross product and the $\gamma$ is the coefficient from the product of $\alpha$ and $\beta$. The out-of-plane bending term is an essential component of nodes' embedding.

**Five-body interactions@I**    The evaluation the five-body@I term could be represented as dihedral angles (depicted in Fig. 2 (d) and Fig. 3 (f)):

$$\left\| \left( \sum_{j \in \mathcal{N}_i} \alpha_j \hat{r}_{ij} \right) \times \left( \sum_{j \in \mathcal{N}_i} \beta_j \hat{r}_{ij} \right) \right\|^2 = \sum_{j_1,j_2,j_3,j_4 \in \mathcal{N}_i} \gamma_{ij_1j_2j_3j_4} \langle \vec{n}_{ij_1j_2}, \vec{n}_{ij_3j_4} \rangle. \tag{7}$$

As noted before, different coefficients $\alpha_j$ and $\beta_j$ are assigned to avoid the term becoming zero and the term provides a node embedding. Moreover, the term encompasses a portion of the description for improper interactions, and further details can be found in the Supplementary Materials.

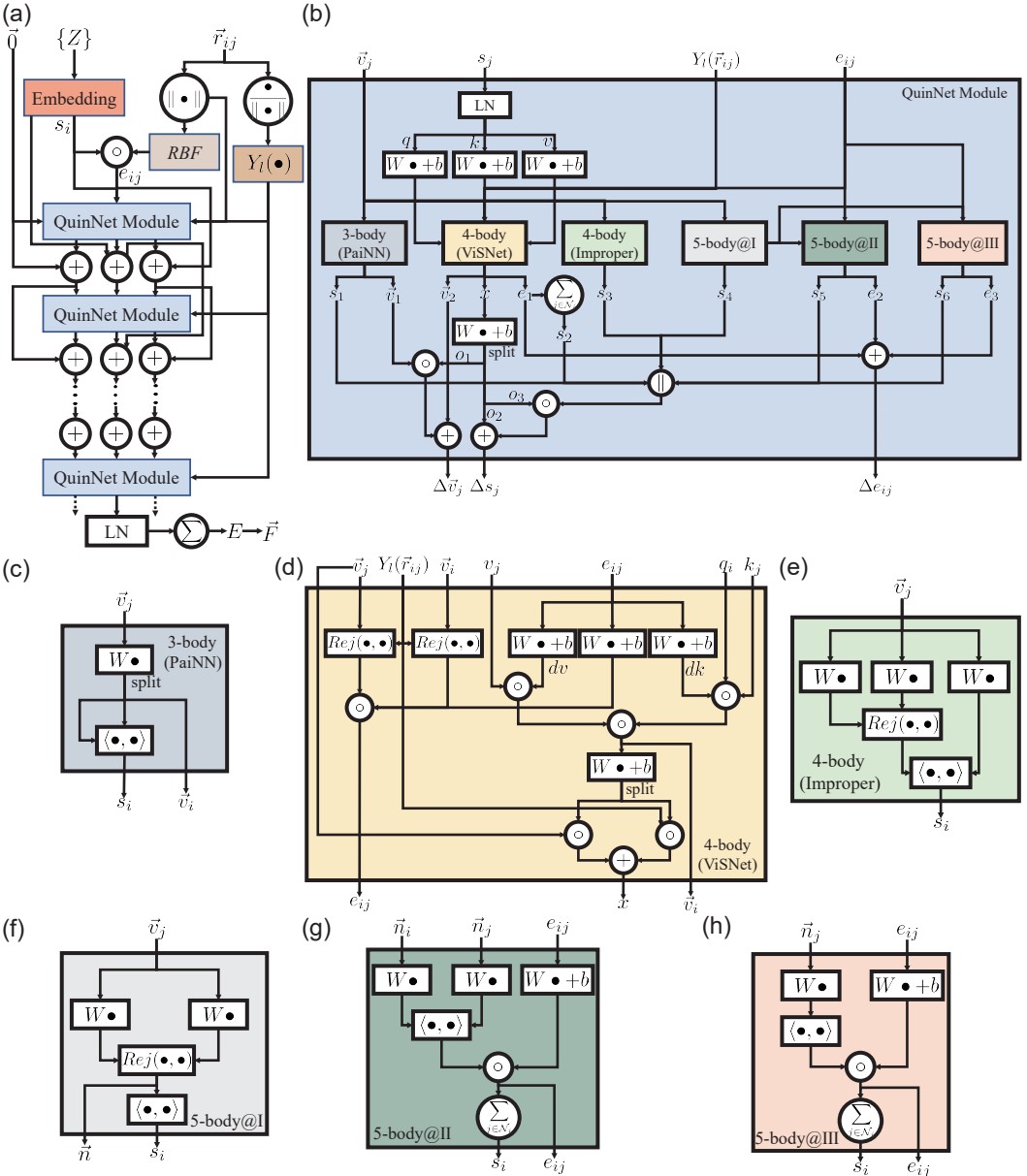

Figure 3: A schematic diagram of QuinNet. (a) The overall architecture of QuinNet, where $\{Z\}$ represents the set of atomic numbers. The initial node features are zero vectors of a fixed dimension. The QuinNet module inputs scalar and vectorial node features $s_i$ and $\vec{v}_i$, radial basis function (RBF) as scalar edge feature $e_{ij}$, and relative position vector $\vec{r}_{ij}$ as vectorial edge feature, and then outputs an updated scalar and vectorial node feature and scalar edge feature for the next layer. The set of spherical harmonic functions, $Y_lm$, is denoted as $Y_l$. The final scalar node features are summed to form the graph feature, and the target prediction is obtained. (b) The QuinNet module consists of six modules that handle (c) 3-body interactions, (d) torsion terms in 4-body interactions, (e) improper terms in 4-body interactions, and (f)-(h) five-body interactions, respectively. The normal vector corresponding to node $j$ is represented by $\vec{n}_j$. In a linear layer, the weights are referred to as $W$, while the term $b$ signifies the bias.

**Five-body interactions@II** To compute the five-body interaction@II term (refer to Fig. 2 (e) and Fig. 3 (g)), we use the following equation to construct hidden states on edges:

$$\left\| \left( \sum_{k \in \mathcal{N}_{j_1}} \alpha_k \hat{r}_{kj_1} \times \sum_{k \in \mathcal{N}_{j_1}} \beta_k \hat{r}_{kj_1} \right) \Big|_{j_1 \in \mathcal{N}_i} \right\|^2 = \left. \sum_{\substack{k_1 \in \mathcal{N}_{j_1}, \\ k_2 \in \mathcal{N}_{j_2}}} \gamma_{j_1 j_2 k_1 k_2} \langle \vec{n}_{ij_1 k_1}, \vec{n}_{ij_2 k_2} \rangle \right|_{j_1, j_2 \in \mathcal{N}_i}, \quad (8)$$

where $\big|_j \in \mathcal{N}_i$ indicates that the values should be chosen according to the source nodes of node $i$.

**Five-body interactions@III** The five-body interaction@III term, as illustrated in Fig. 2 (f) and Fig. 3 (h), is computed as follows,

$$\left. \left( \sum_{j \in \mathcal{N}_i} \alpha_j \hat{r}_{ij} \times \sum_{j \in \mathcal{N}_i} \beta_j \hat{r}_{ij} \right) \cdot \left( \sum_{k \in \mathcal{N}_j} \alpha_k \hat{r}_{jk} \times \sum_{k \in \mathcal{N}_j} \beta_k \hat{r}_{jk} \right) \right|_{j \in \mathcal{N}_i} = \left. \left. \sum_{\substack{j_1, j_2 \in \mathcal{N}_i, \\ k_1, k_2 \in \mathcal{N}_j}} \gamma_{ij_1 j_2 k_1 k_2} \langle \vec{n}_{ij_1 j_2}, \vec{n}_{jk_1 k_2} \rangle \right|_{j \in \mathcal{N}_i} \right|_{j \in \mathcal{N}_i},$$

(9)

which constructs the representation of edges.

**Higher order cosine series** Since it is beneficial to include higher order cosine series terms in the force field [41, 42], we utilize the vector addition theorem of spherical harmonic functions. Although all the angular calculations in Eq. 4 - Eq. 9 only consider one order of cosine terms, the higher-order cosine terms could be incorporated easily by using the following equation:

$$P_l(\cos \theta) = \frac{4\pi}{2l+1} \sum_{m=-l}^{l} Y_{lm}^*(\hat{u}) Y_{lm}(\hat{v}), \quad (10)$$

where $\theta$ denotes the angle between two unit vectors $\hat{u}$ and $\hat{v}$, and $P_l$ and $Y_l$ represent Legendre polynomials and spherical harmonic functions of $l$-th order, respectively. $Y_l^*$ is the complex conjugate of $Y_l$. As $l$ increases, higher-order cosine terms are incorporated into the calculation, enabling us to improve the accuracy of our model and make it suitable for a broader range of applications. Additional proof can be found in the Supplementary Materials.

### 4.3 Complexity analysis

Inspired by the design of PaiNN [15], QuinNet incorporates many-body interactions by calculating relevant physical quantities during the message passing layer, eliminating the need for calculations beforehand. The operations within the QuinNet module can be decomposed into cross product and inner product, which are independent of the number of nodes' neighbors. As a result, the computational complexity of each many-body interaction is significantly reduced to $\mathcal{O}(|\mathcal{N}|)$, where $|\mathcal{N}|$ represents the number of neighbors for each node. This approach allows QuinNet to efficiently model complex interactions, enhancing its overall performance. A more comprehensive analysis refers to the Supplementary Materials.

## 5 Results

We conduct comprehensive evaluations on different datasets to demonstrate the effectiveness of the QuinNet model. First, we benchmark our model on the public MD17 [2] and revised MD17 [44] small molecular datasets, where the impact of five-body interactions is relatively weak. Despite this, we show that the QuinNet model shows comparable accuracy with the state-of-the-art models in these datasets. Moreover, QuinNet has also been benchmarked on the QM9 dataset [45, 46], the details of which can be found in the Supplementary Materials. Furthermore, we evaluate the performance of QuinNet on the larger molecular systems of MD22 [47] and Chignolin datasets [48]. These complex systems are known to exhibit a more pronounced influence of five-body interactions. Our evaluation demonstrates that QuinNet accurately models these interactions and achieves higher accuracy in energy and force prediction compared to other models.

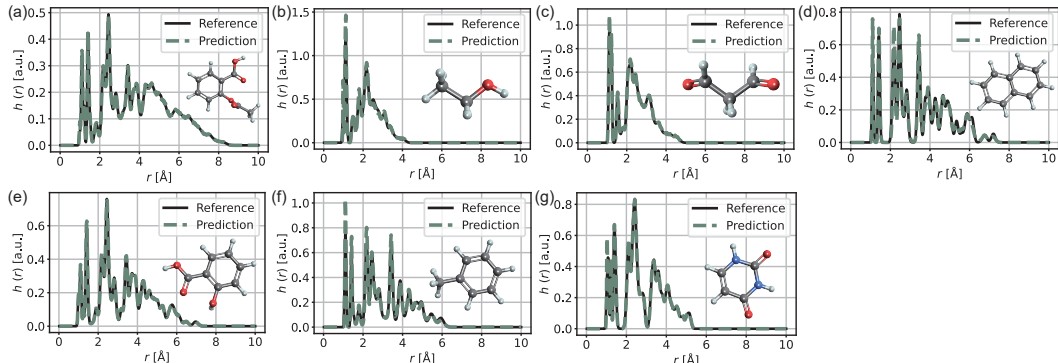

Figure 4: The distribution of interatomic distances $h(r)$ for (a) aspirin, (b) ethanol, (c) malonaldehyde, (d) naphthalene, (e) salicylic acid, (f) toluene, and (g) uracil in MD17 dataset. The insets display the ball-and-stick representations of these molecules.

## 5.1 MD17 Dataset and MD Simulation

The MD17 dataset [2] contains a diverse range of sizes, spanning from 150k to nearly 1M conformational geometries. Each trajectory was computed at a temperature of 500 K with a resolution of 0.5 fs. The total energy and force labels for each dataset were computed using the PBE+vdW-TS electronic structure method [49, 50]. In Table 1, we present the mean absolute errors (MAEs) of QuinNet for 7 molecules in the MD17 dataset, along with comparisons to nine other models. QuinNet outperforms other tested models in nearly 57.14% of tasks, especially the performance on forces. Additionally, we perform MD simulations using trained QuinNets as force fields and plot the distribution of interatomic distances $h(r)$ for these 7 molecules in Fig. 4. Further details regarding additional settings can be found in the Supplementary Materials.

Table 1: Comparison of the MAEs between several benchmarked models and QuinNet trained on MD17 dataset using 950 training samples and 50 validation samples (energies in kcal/mol and forces in kcal/(mol·Å)). The lowest values are highlighted in bold.

| | | SchNet [11, 12] | DimeNet [13] | PaiNN [15] | SpookyNet [51] | ET [16] | GemNet [17] | NequIP ($l$=3) [7] | SO3KRATES [52] | ViSNet [20] | QuinNet |
|---|---|---|---|---|---|---|---|---|---|---|---|
| Aspirin | Energy | 0.37 | 0.204 | 0.167 | 0.151 | 0.123 | - | 0.131 | 0.139 | **0.116** | 0.119 |
| | Force | 1.35 | 0.499 | 0.338 | 0.258 | 0.253 | 0.217 | 0.184 | 0.236 | 0.155 | **0.145** |
| Ethanol | Energy | 0.08 | 0.064 | 0.064 | 0.052 | 0.052 | - | 0.051 | 0.052 | 0.051 | **0.050** |
| | Force | 0.39 | 0.230 | 0.224 | 0.094 | 0.109 | 0.085 | 0.071 | 0.096 | **0.060** | **0.060** |
| Malonaldehyde | Energy | 0.13 | 0.104 | 0.091 | 0.079 | 0.077 | - | 0.076 | 0.077 | **0.075** | 0.078 |
| | Force | 0.66 | 0.383 | 0.319 | 0.167 | 0.169 | 0.155 | 0.129 | 0.147 | 0.100 | **0.097** |
| Naphthalene | Energy | 0.16 | 0.122 | 0.116 | 0.116 | **0.085** | - | 0.113 | 0.115 | **0.085** | 0.101 |
| | Force | 0.58 | 0.215 | 0.077 | 0.089 | 0.061 | 0.051 | **0.039** | 0.074 | 0.039 | 0.039 |
| Salicylic acid | Energy | 0.20 | 0.134 | 0.116 | 0.114 | 0.093 | - | 0.106 | 0.016 | **0.092** | 0.101 |
| | Force | 0.85 | 0.374 | 0.195 | 0.180 | 0.129 | 0.125 | 0.090 | 0.145 | 0.084 | **0.080** |
| Toluene | Energy | 0.12 | 0.102 | 0.095 | 0.094 | **0.074** | - | 0.092 | 0.095 | **0.074** | 0.080 |
| | Force | 0.57 | 0.216 | 0.094 | 0.087 | 0.067 | 0.060 | 0.046 | 0.073 | **0.039** | **0.039** |
| Uracil | Energy | 0.14 | 0.115 | 0.106 | 0.105 | **0.095** | - | 0.104 | 0.103 | **0.095** | 0.096 |
| | Force | 0.56 | 0.301 | 0.139 | 0.119 | 0.095 | 0.097 | 0.076 | 0.111 | **0.062** | **0.062** |

## 5.2 Revised MD17 Dataset

Christensen and von Lilienfeld discovered that the energy of the original MD17 dataset was noisy [44]. Therefore, 100,000 structures of each molecule were selected from the original dataset to create the revised MD17 dataset, and the energies and forces were recalculated at the PBE/def2-SVP level of theory using a very tight SCF convergence and a very dense DFT integration grid [49, 53, 54]. As shown in Table 2, QuinNet matches the performance of popular test models on the revised MD17 dataset. Supplementary Materials provide additional information on the relevant settings.

## 5.3 MD22 Dataset

The MD22 dataset [47] is a recently introduced collection of MD trajectories, which encompasses four major classes of biomolecules and supramolecules, ranging from a small peptide consisting

Table 2: Comparison of the MAEs between several benchmarked models and QuinNet trained on revised MD17 dataset using 950 training samples and 50 validation samples (energies in kcal/mol and forces in kcal/(mol·Å)). The lowest values are shown in bold.

| | | UNiTE [55] | GemNet (T/Q) [17] | NequIP ($l$=3) [7] | MACE [37] | Allegro [36] | BOTNet | ViSNet [20] | QuinNet |
|---|---|---|---|---|---|---|---|---|---|
| Aspirin | Energy | 0.055 | - | 0.0530 | 0.0507 | 0.0530 | 0.0530 | **0.0445** | 0.0486 |
| | Force | 0.175 | 0.2191 | 0.1891 | 0.1522 | 0.1684 | 0.1960 | 0.1520 | **0.1429** |
| Azobenzene | Energy | 0.025 | - | 0.0161 | 0.0277 | 0.0277 | 0.0161 | **0.0156** | 0.0394 |
| | Force | 0.097 | - | 0.0669 | 0.0692 | 0.0600 | 0.0761 | 0.0585 | **0.0513** |
| Benzene | Energy | 0.002 | - | 0.0009 | 0.0092 | 0.0069 | **0.0007** | **0.0007** | 0.0096 |
| | Force | 0.017 | 0.0115 | 0.0069 | 0.0069 | **0.0046** | 0.0069 | 0.0056 | 0.0047 |
| Ethanol | Energy | 0.014 | - | 0.0092 | **0.0032** | 0.0092 | 0.0092 | 0.0078 | 0.0096 |
| | Force | 0.085 | 0.083 | 0.0646 | **0.0484** | **0.0484** | 0.0738 | 0.0522 | 0.0516 |
| Malonaldehyde | Energy | 0.025 | - | 0.0184 | 0.0185 | 0.0138 | 0.0185 | **0.0132** | 0.0168 |
| | Force | 0.152 | 0.1522 | 0.01176 | 0.0946 | **0.0830** | 0.1338 | 0.0893 | 0.0875 |
| Naphthalene | Energy | 0.011 | - | **0.0046** | 0.1153 | **0.0046** | **0.0046** | 0.0057 | 0.0174 |
| | Force | 0.060 | 0.0438 | 0.0300 | 0.0369 | **0.0208** | 0.0415 | 0.0291 | 0.0242 |
| Paracetamol | Energy | 0.044 | - | 0.0323 | 0.0300 | 0.0346 | 0.0300 | **0.0258** | 0.0362 |
| | Force | 0.164 | - | 0.1361 | 0.1107 | 0.1130 | 0.1338 | 0.1029 | **0.0979** |
| Salicylic acid | Energy | 0.017 | - | **0.0161** | 0.0208 | 0.0208 | 0.0185 | **0.0161** | 0.033 |
| | Force | 0.088 | 0.1222 | 0.0922 | 0.0715 | **0.0669** | 0.0992 | 0.0795 | 0.0771 |
| Toluene | Energy | 0.010 | - | 0.0069 | 0.0115 | 0.0092 | 0.0069 | **0.0059** | 0.0139 |
| | Force | 0.058 | 0.0507 | 0.0369 | 0.0350 | 0.0415 | 0.0438 | 0.0264 | **0.0244** |
| Uracil | Energy | 0.013 | - | 0.0092 | 0.0115 | 0.0138 | 0.0092 | **0.0069** | 0.0149 |
| | Force | 0.088 | 0.0876 | 0.0669 | 0.0484 | **0.0415** | 0.0738 | 0.0495 | 0.0487 |

Table 3: Comparison of the MAEs between several benchmarked models and the lowest values are marked in bold (energies in kcal/mol per atom and forces in kcal/(mol·Å)). Furthermore, the data splitting of the training set and validation set for training QuinNet on MD22 dataset is shown.

| | # Train/Val | | sGDML [47] | ViSNet-LSRM [56] | ViSNet [20, 56] | MACE (3Å) [57] | MACE (6Å) [57] | MACE (5Å) [57] | QuinNet |
|---|---|---|---|---|---|---|---|---|---|
| Ac-Ala3-NHMe | 5500/500 | Energy | 0.0093 | **0.0016** | 0.0019 | 0.0140 | 0.0080 | 0.0015 | 0.0020 |
| | | Force | 0.79 | 0.0942 | 0.0972 | 0.1753 | 0.3920 | 0.0876 | **0.0681** |
| DHA (docosahexaenoic acid) | 7500/500 | Energy | 0.023 | **0.0016** | 0.0027 | 0.0103 | 0.0092 | 0.0024 | 0.0021 |
| | | Force | 0.75 | 0.0598 | 0.0668 | 0.1430 | 0.5419 | 0.0646 | **0.0515** |
| Stachyose | 7500/500 | Energy | 0.046 | **0.0012** | 0.0015 | 0.0058 | 0.0082 | 0.0014 | 0.0026 |
| | | Force | 0.68 | 0.0767 | 0.0869 | 0.1568 | 0.6226 | 0.0876 | **0.0543** |
| AT-AT | 2500/500 | Energy | 0.012 | **0.0013** | 0.0028 | 0.0208 | 0.0036 | 0.0018 | 0.0024 |
| | | Force | 0.69 | 0.0781 | 0.1070 | 0.3067 | 0.3436 | 0.0992 | **0.0687** |
| AT-AT-CG-CG | 1500/500 | Energy | 0.012 | **0.0010** | 0.0017 | 0.0139 | 0.0038 | 0.0013 | 0.0032 |
| | | Force | 0.70 | **0.1064** | 0.1563 | 0.3759 | 0.4635 | 0.1153 | 0.1273 |
| Buckyball catcher | 550/50 | Energy | 0.0079 | **0.0029** | 0.0030 | 0.0110 | 0.0039 | 0.0033 | 0.0038 |
| | | Force | 0.68 | 0.1026 | 0.1335 | 0.3021 | 0.5120 | **0.0853** | 0.1091 |
| Double-walled nanotube | 750/50 | Energy | 0.0108 | 0.0049 | **0.0028** | 0.0048 | 0.0053 | 0.0045 | 0.0049 |
| | | Force | 0.52 | 0.3391 | 0.3959 | 0.4128 | 0.9132 | 0.2767 | **0.2473** |

of 42 atoms to a double-walled nanotube containing 370 atoms. The trajectories were sampled at temperatures between 400 and 500 K at a resolution of 1 fs, and the corresponding potential energy and atomic forces were calculated using the PBE+MBD [49, 50] level of theory. Table 3 presents the MAEs of the QuinNet model for seven systems within the MD22 dataset, along with a comparison to six benchmark models. Our results show that QuinNet outperforms the vast majority of benchmark models in predicting atomic forces, and since the MD22 dataset claims that long-range interactions are crucial, this could be the reason why the ViSNet-LSRM model that incorporates long-range interactions and the models with chosen cutoff radius have more accurate energy predictions. Further details on the relevant settings refer to the Supplementary Materials.

## 5.4 Ablation study on Chignolin Dataset

The Chignolin, consisting of 166 atoms, serves as the simplest artificial protein for folding and unfolding studies. The dataset includes a diverse set of folding and unfolding states of Chignolin, comprising a total of 9,543 representative conformations [48]. As Wang et al.[25] noted that incorporating five-body interactions could enhance the performance of models on Chignolin, we performed an ablation study to evaluate the impact of various many-body interactions. As shown in Fig. 5, the explicit inclusion of five-body interactions up to part II led to a significant decrease in MAEs. It is worth noting that the energy errors exhibit some fluctuations when introducing five-body interactions up to part I. Considering that the five-body interactions up to part I to some extent describe the four-body improper terms, as demonstrated in the Supplementary Materials, we speculate that these two different representations of the same interaction may cause antagonism. To test this, we

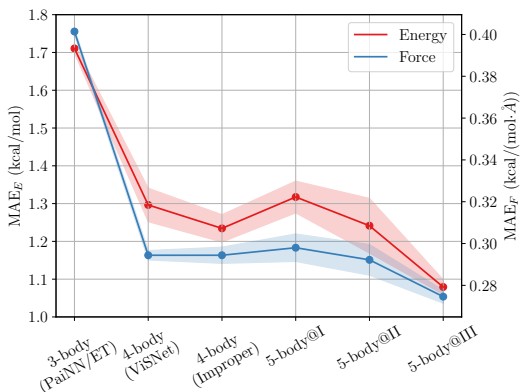

Figure 5: The ablation study on Chignolin dataset.

removed the four-body (improper) terms and retained only the five-body interactions up to part I for training on the Chignolin dataset. The resulting energy MAE is $1.2906 \pm 0.0267$ kcal/mol, and the force MAE is $0.2974 \pm 0.002417$ kcal/(mol·Å). These values are lower than the case where both four-body (improper) and five-body interactions up to part I coexist, thus confirming our hypothesis. These results demonstrate the importance of incorporating five-body interactions for accurately modeling the folding and unfolding conformations of Chignolin. Comparisons of the MAEs between different models and the detailed values of the ablation study refer to the Supplementary Materials.

## 6  Conclusion

In this work, we propose the QuinNet architecture, which efficiently incorporates many-body interactions up to whole five-body in graph neural networks for molecular dynamics simulations. Our experiments on several public datasets, including MD17, revised MD17, MD22, and Chignolin, demonstrate that QuinNet achieves high accuracy without significantly increasing computational complexity. Notably, our ablation study on Chignolin highlights the significance of five-body interactions in accurately modeling complex bio-molecular systems.

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
