# Supplementary Material:
# QuinNet: Efficiently Incorporating Quintuple Interactions into Geometric Deep Learning force Fields

**Zun Wang**
Microsoft Research AI4Science
Beijing, China, 100084
zunwang@microsoft.com

**Guoqing Liu**
Microsoft Research AI4Science
Beijing, China, 100084

**Yichi Zhou**
Microsoft Research AI4Science
Beijing, China, 100084

**Tong Wang**
Microsoft Research AI4Science
Beijing, China, 100084

**Bin Shao**
Microsoft Research AI4Science
Beijing, China, 100084
binshao@microsoft.com

## 1 Supplementary Material

### 1.1 Proof of equivariance

As all the many-body interactions in the QuinNet are calculated based on inner product and cross product, the proof for the equivariance of these modules is equivalent to prove the equivariance of inner product and cross product. Let $\mathbf{R}$ be a $3\times3$ rotation matrix, i.e. $\det \mathbf{R} = 1$ and $\mathbf{R}^{-1} = \mathbf{R}^T$, for all vectors $\vec{u}, \vec{v} \in \mathbb{R}^3$,

$$
\begin{aligned}
(\mathbf{R}\vec{u}) \cdot (\mathbf{R}\vec{v}) &= (\mathbf{R}\vec{u})_i (\mathbf{R}\vec{v})_i \\
&= R_{ij} u_j R_{ik} v_k \\
&= R_{ij} R_{ik} u_j v_k \\
&= (\mathbf{R}^T)_{ji} \mathbf{R}_{ik} u_j v_k \\
&= (\mathbf{R}^T \mathbf{R})_{jk} u_j v_k \\
&= \delta_{jk} u_j v_k \\
&= u_j v_j \\
&= \vec{u} \cdot \vec{v},
\end{aligned}
\tag{1}
$$

37th Conference on Neural Information Processing Systems (NeurIPS 2023).

Table S 1: Comparison of the MAEs on Chignolin dataset and the lowest values are marked in bold (energies in kcal/mol and forces in kcal/(mol·Å)).

| | | ViSNet-LSRM | 3-body (ET) | 4-body (ViSNet) | 4-body (improper) | 5-body@I | 5-body@II | 5-body (QuinNet) | QuinNet (6 Layer) |
|---|---|---|---|---|---|---|---|---|---|
| Chignolin | Energy | 1.227 | 1.711± 0.012 | 1.296± 0.044 | 1.234± 0.036 | 1.317± 0.042 | 1.241± 0.072 | 1.079± 0.019 | **1.036** |
| | Force | 0.2778 | 0.4014± 0.0015 | 0.2944± 0.0022 | 0.2944± 0.0039 | 0.2980± 0.0066 | 0.2922± 0.0073 | 0.2747± 0.0030 | **0.2665** |

where $\delta_{jk}$ is the Kronecker delta symbol and the Einstein summation convention is used. To prove the equivariance of cross product, the Levi-Civita permutation symbol $\epsilon$ would be used,

$$
\begin{aligned}
[(\mathbf{R}\vec{u}) \times (\mathbf{R}\vec{v})]^k &= \epsilon^{imk} R_{ij} R_{mn} u^j v^n \\
&= \epsilon^{iml} \delta_{kl} R_{ij} R_{mn} u^j v^n \\
&= \epsilon^{iml} R_{kr} R_{lr} R_{ij} R_{mn} u^j v^n \\
&= \epsilon^{jnr} \det \mathbf{R} R_{kr} u^j v^n \\
&= R_{kr} \epsilon^{jnr} u^j v^n \\
&= R_{kr} (\vec{u} \times \vec{v})^r \\
&= \mathbf{R} \cdot (\vec{u} \times \vec{v})^k.
\end{aligned}
\tag{2}
$$

## 1.2 Proof of higher order cosine series

The Legendre polynomials could be expressed as

$$
P_l(\cos\theta) = 2^l \sum_{k=0}^{l} \cos^k \theta \binom{l}{k} \binom{\frac{l+k-1}{2}}{l}.
\tag{3}
$$

Incorporating higher order cosine series into the QuinNet model is necessary in certain cases. These series can be represented as $\cos(n\theta)$ and can be expanded as a linear combination of $\cos^k \theta$ according to two-fold duplication formula, i.e. $\cos 2\theta = 2\cos^2 \theta - 1$. Therefore, the vector addition theorem of spherical harmonic functions can be used to incorporate these higher order cosine series into the model.

## 1.3 Additional results on Chignolin

Table S 1 reports the MAEs of the benchmarked model, along with the results of the ablation study. It should be noted that in the ablation study, the benchmarked models, which includes only 3-body interactions, and incorporates 4-body interactions, are ET [1] and ViSNet [2] model respectively. Furthermore, the results of the 6-layer QuinNet model are presented in the table.

## 1.4 Results on QM9 dataset

The QM9 dataset [3, 4] encompasses computed geometric, energetic, electronic, and thermodynamic properties of 134k stable small organic molecules, which include carbon, hydrogen, oxygen, nitrogen, and fluorine, ascertained at the B3LYP/6-31G (2df, p) level of quantum chemistry. This dataset offers valuable quantum chemical insights into the chemical space of small organic molecules and is widely acknowledged as a benchmark for calibrating, analyzing, and evaluating new methods in this area. As a result, we trained QuinNet on 110k molecules and validated it on a further 10k molecules. Table 2 presents the mean absolute errors (MAEs) of QuinNet for 12 tasks in the QM9 dataset, compared to four other models. Despite the QM9 dataset being a small molecular dataset where the influence of five-body interactions is relatively weak, QuinNet's MAEs are in line with these baselines. It is worth noting that the gap is computed directly from the predicted HOMO and LUMO values.

## 1.5 Further Complexity Analysis

To further highlight the efficiency of the QuinNet model, Table S 3 presents a comparison of time complexities for handling many-body interactions between QuinNet and the empirical force field. In the table, $N$ and $N_b$ denote the number of atoms and the number of neighbors, respectively. Note that the number of atoms $N$ is ignored in the complexity analysis of QuinNet. Additionally, we

Table S 2: Comparison of the MAEs on QM9 dataset and the lowest values are marked in bold.

| | unit | Allegro [5] | Equiformer [6] | ViSNet [2] | QuinNet |
|---|---|---|---|---|---|
| $\mu$ | $D$ | - | 0.014 | **0.010** | 0.771 |
| $\alpha$ | $a_0^3$ | - | 0.056 | **0.041** | 0.047 |
| HOMO | meV | - | **17** | **17.3** | 20.4 |
| LUMO | meV | - | 16 | **14.8** | 17.6 |
| gap | meV | - | 33 | 31.7 | **28.2** |
| $R^2$ | $a_0^2$ | - | 0.227 | **0.030** | 0.194 |
| ZPVE | meV | - | 1.32 | 1.56 | **1.26** |
| $U_0$ | meV | 4.7 | 10 | **4.23** | 7.6 |
| $U$ | meV | 4.4 | 11 | **4.25** | 8.4 |
| $H$ | meV | **4.4** | 10 | 4.52 | 7.8 |
| $G$ | meV | **5.7** | 10 | 5.86 | 8.5 |
| $C_v$ | $\frac{kcal}{mol \cdot K}$ | - | 0.025 | **0.023** | 0.024 |

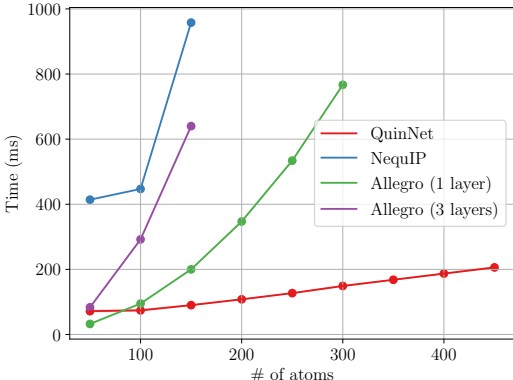

Fig. S 1: The inference time of different models with a single 32GB V100 GPU card.

evaluate the inference time of different models with a single 32GB V100 GPU card (Fig. S1). As the system size increases, the NequIP and Allgro models encounter out-of-memory issues. In general, the inference time of QuinNet is lower compared to the other benchmarked models.

## 1.6 Overlap between many-body interactions

The QuinNet model is designed to effectively capture all possible five-body interactions, which have some overlaps with other many-body interactions actually. Specifically, as shown in Fig. S2 (a), we illustrate the topology of five-body@I interactions, which includes four neighboring nodes, namely $j_1$, $j_2$, $j_3$, and $j_4$, with respect to the central node $i$. However, when $j_2 = j_4$ (Fig. S2 (b)), the topology reduces to a four-body interaction similar to an improper torsion interaction. Through this transformation, the five-body@I term characterizes the improper term associated with dihedral angles. The QuinNet model captures all five-body interactions, making it a versatile and comprehensive tool for modeling complex molecular systems.

## 1.7 Settings of experiments

The loss function for training QuinNet model is the weighted summation of mean square errors of energy and forces,

$$L = \alpha L_E + \beta L_F = \frac{\alpha}{N} \sum_i^N (E_i - \hat{E}_i)^2 + \frac{\beta}{3N_a} \sum_{\substack{i, \\ \sigma=x,y,z}}^{N_a} (F_{i\sigma} - \hat{F}_{i\sigma})^2, \tag{4}$$

where $N$ and $N_a$ are batch size and the number of atoms, respectively. AdamW optimizer [7] was adopted. QuinNet model was trained on 32G Nvidia Tesla V100 GPUs for MD17, revised MD17

Table S 3: Comparison of time complexities for handling many-body interactions between QuinNet and the empirical force field.

| N-body | Empirical force field | Pseudocode of QuinNet | Time Complexity Empirical force field | QuinNet |
|---|---|---|---|---|
| 3-body | For each atom $i$, choose two neighbor atoms $j$ and $k$ to calculate angles. | 1: $m_i = 0$
2: **for** $j \in \mathcal{N}_i$ **do**
3: $\quad m_i += \vec{r}_{ij}$
4: **end for**
5: $h_i = m_i^2$ | $NC_{N_b}^2 \sim \mathcal{O}(NN_b^2)$ | $\mathcal{O}(N_b)$ |
| 4-body (torsion) | For each atom $i$, choose two neighbor atoms $j$ and $k$ firstly, then choose one neighbor atom $l$ of atom $j$ to calculate dihedral angles. | 1: $h_{ij,1}, h_{ij,2} = 0, 0$
2: **for** $k_1 \in \mathcal{N}_i$ **do**
3: $\quad h_{ij,1} += \vec{r}_{ik_1} \times \vec{r}_{ij}$
4: **end for**
5: **for** $k_2 \in \mathcal{N}_j$ **do**
6: $\quad h_{ij,2} += -\vec{r}_{jk_2} \times \vec{r}_{ij}$
7: **end for**
8: $h_{ij} = h_{ij,1} \cdot h_{ij,2}$ | $NC_{N_b}^2 C_{N_b}^1 \sim \mathcal{O}(NN_b^3)$ | $\mathcal{O}(N_b)$ |
| 4-body (improper) | For each atom $i$, choose three neighbor atoms $j$, $k$, and $l$ to calculate angles. | 1: $m_i, h_{i,1}, h_{i,2} = 0, 0, 0$
2: **for** $j \in \mathcal{N}_i$ **do**
3: $\quad m_i += \vec{r}_{ij}$
4: $\quad h_{i,1} += \alpha_j \vec{r}_{ij}$
5: $\quad h_{i,2} += \beta_j \vec{r}_{ij}$
6: **end for**
7: $h_i = m_i \cdot (h_{i,1} \times h_{i,2})$ | $NC_{N_b}^3 \sim \mathcal{O}(NN_b^3)$ | $\mathcal{O}(N_b)$ |
| 5-body@I | For each atom $i$, choose four neighbor atoms $j$, $k$, $l$, and $m$ to calculate dihedral angles. | 1: $h_{i,1}, h_{i,2} = 0, 0$
2: **for** $j \in \mathcal{N}_i$ **do**
3: $\quad h_{i,1} += \alpha_j \vec{r}_{ij}$
4: $\quad h_{i,2} += \beta_j \vec{r}_{ij}$
5: **end for**
6: $h_i = (h_{i,1} \times h_{i,2})^2$ | $NC_{N_b}^4 \sim \mathcal{O}(NN_b^4)$ | $\mathcal{O}(N_b)$ |
| 5-body@II | For each atom $i$, choose two neighbor atoms $j_1$ and $j_2$, then choose one neighbor atom $k$ for each atom $j$ to calculate dihedral angles. | 1: $h_{j,1}, h_{j,2}, h_i = 0, 0, 0$
2: **for** $k \in \mathcal{N}_j$ **do**
3: $\quad h_{j,1} += \alpha_k \vec{r}_{kj}$
4: $\quad h_{j,2} += \beta_k \vec{r}_{ij}$
5: **end for**
6: **for** $j \in \mathcal{N}_i$ **do**
7: $\quad h_i += h_{j,1} \times h_{j,2}$
8: **end for**
9: $h_i = h_i^2$ | $NC_{N_b}^2 C_{N_b}^1 C_{N_b}^1 \sim \mathcal{O}(NN_b^4)$ | $\mathcal{O}(N_b)$ |
| 5-body@III | For each atom $i$, choose three neighbor atoms $j_1$, $j_2$, and $j_3$, then choose one neighbor atom $k$ for one of the thee neighbor atoms of $i$ to calculate dihedral angles. | 1: $h_{j,1}, h_{j,2} = 0, 0$
2: **for** $k \in \mathcal{N}_j$ **do**
3: $\quad h_{j,1} += \alpha_k \vec{r}_{kj}$
4: $\quad h_{j,2} += \beta_k \vec{r}_{kj}$
5: **end for**
6: $h_j = h_{j,1} \times h_{j,2}$
7: **for** $j \in \mathcal{N}_i$ **do**
8: $\quad h_{ij} += h_i h_j$
9: **end for** | $NC_{N_b}^3 C_{N_b}^1 \sim \mathcal{O}(NN_b^4)$ | $\mathcal{O}(N_b)$ |

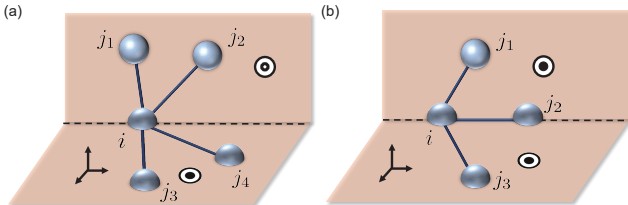

Fig. S 2: A comparison between five-body and many-body interactions. (a) five-body interactions@I, which involves four neighboring nodes, can overlap with (b) the improper term in four-body interactions, resulting in a four-body interaction topology.

Table S 4: Hyperparameters of QuinNet for different datasets.

| | MD17 | revised MD17 | MD22 | Chignolin | QM9 |
|---|---|---|---|---|---|
| Energy/force weights | 0.01, 0.99 | 0.01, 0.99 | 0.01, 0.99 | 0.01, 0.99 | - |
| Energy/force ema | 0.05, 1.0 | 0.05, 1.0 | 0.05, 1.0 | 0.05, 1.0 | - |
| Cutoff (Å) | 4.0, 5.0 | 4.0, 5.0 | 4.0, 5.0 | 5.0 | 5.0 |
| # layers | 5 | 5 | 5 | 5 | 5 |
| # neurons | 256 | 256 | 256 | 256 | 256 |
| Batch size | 4 | 2, 4 | 2, 4 | 8 | 16 |
| Learning rate (LR) | 2e-4, 4e-4 | 2e-4, 4e-4 | 2e-4, 4e-4 | 2e-4 | 2e-4, 3e-4, 4e-4, 5e-4 |
| LR decay factors | 0.8 | 0.8 | 0.8 | 0.8 | 0.8 |

and Chignolin dataset. For MD22 dataset, the model was trained on a single 80G Nvidia Tesla A100 GPU. Furthermore, detailed settings of hyperparameters are summarized in the Table S 4.

Moreover, to demonstrate QuinNet's performance, we performed molecular dynamics (MD) simulations of seven small molecules from the MD17 dataset. For each model and molecule, simulations began with the initial frame configurations and were performed over 300 ps. We employed a 0.5 fs time step and maintained the temperature at 500 K using a Nosé-Hoover thermostat. The distribution of interatomic distances, h(r), was calculated as the ensemble average of distance statistics within the trajectories. The code for the simulation was implemented using the ASE [8] Python package and adapted from Ref. [9].