# OpenReview forum: "Efficiently incorporating quintuple interactions into geometric deep learning force fields"
_NeurIPS.cc/2023/Conference — NeurIPS 2023 poster_

### Official Review · Reviewer_um1j · 2023-06-28

**Soundness:** 4 excellent
**Presentation:** 3 good
**Contribution:** 3 good
**Rating:** 7
**Confidence:** 4

**Summary:**

The paper introduces a new method for molecular modeling, QuinNet, which incorporates five-body interactions using only dihedral angles. The authors first introduce relevant concepts related to machine learning force fields and related work in the field related to a variety of equivariant models. Next, the paper describes pertinent definitions of force fields, group equivariance, and methods for calculating empirical force fields. In the methods section, the authors describe their approach for integrating five-body terms into the architecture of QuiNet using only dihedral angles and incorporating model designs from prior work (PaiNN for 3-body interactions, ViSNet for 4-body interactions) and new definitions for different topologies of 5-body interactions. In addition to the architectural description, the authors provide relevant mathematical formulations and a complexity analysis. In their results, the authors showcase QuiNets performance on a low (MD17) and high complexity (MD-22) dataset in terms of energy and force modeling, including an ablation for different body terms in Figure 5.

**Strengths:**

The paper has the following strengths:
* Originality: The proposed architecture incorporates relevant terms for molecular modeling that are physically relevant, but have not been incorporated before.
* Quality: The method and experimental design showcase relevant cases for applying GNN models for molecular modeling with the idea behind the architecture being well-motivated.
* Clarity: The paper presents a cohesive formulation of their method, both in figures and mathematics, and experiment descriptions with relevant takeaways.
* Significance: The proposed architecture shows improved modeling performance, especially in forces, and provides a potential framework for incorporating physical interactions into GNNs.

**Weaknesses:**

The paper could be improved by the following:
* Providing a clear and concise discussion of limitations. [Quality, Significance]
* Adding more context for the results in Figure 4. The MD simulations are only briefly described in Section 5.1, which is on a different page then the figure and easy to miss. [Clarity]
* A description of the case in which a greater set of many-body interactions is beneficial. This is briefly mentioned in the discussion between MD17 and MD22, but it would be good to put in greater context in terms of the experimental results and could serve as part of the conclusion. [Clarity]

**Questions:**

* Could you provide additional details on the limitations of QuinNet? E.g. Is it limited to modeling mainly molecular systems? What sizes of molecules do you think QuinNet can be effective in and why?
* Do you have data that supports your compute complexity analysis compared to other methods? If so, what kind of speedup do you generally find, if any?

**Limitations:**

The authors do not provide a discussion on limitations, which I raised as a weakness. I would like to see a discussion of limitations in future versions and/or during the discussion period.

---

> ### Author Rebuttal · Authors · 2023-08-10
>
> We thank the reviewer for his/her comments and will address each point in our response accordingly.
>
> ### Weakness 1:
> * As the experimental results indicate, five-body interactions do not have a substantial impact on small molecules. To demonstrate the significance of these interactions, it is essential to examine larger molecular systems.
>
> ### Weakeness 2:
> * We apologize for not providing detailed information on the MD settings. Simulations were conducted for each model/molecule, covering a duration of 300 ps and starting from the initial frame configurations. With a 0.5 fs time step and a maintained temperature of 500 K, the simulations were controlled using a Nosé-Hoover thermostat. The distribution of interatomic distances, h(r), was computed as the ensemble average of distance statistics within the trajectories. We will include these relevant details in the Supplementary Materials section.
>
> ### Weakness 3:
> * As evidenced by the experimental results, higher-order many-body interactions become more remarkable for larger molecules. We appreciate the reviewer's suggestion and will incorporate additional discussion on this topic in the conclusion section.
>
> ### Questions 1:
> * As previously mentioned in response to weakness 1, our experimental results indicate that five-body interactions do not have a significant impact on small molecules. To demonstrate the importance of five-body interactions, larger molecular systems should be investigated. While our current experiments primarily focus on molecular systems, we plan to extend our methods to more complex systems, such as periodic structures. As we have stated, to highlight the significance of five-body interactions, larger systems must be examined, with the largest system in our experiments containing 370 atoms. The results show an improvement when incorporating five-body interactions. Based on the findings from the MD22 and Chignolin dataset, molecules with nearly 100 atoms exhibit improvements when five-body interactions are included.
>
> ### Questions 2:
> * Time Complexity section in the official comment illustrates the complexity of calculating related physical quantities explicitly, as well as the calculations performed in QuinNet. The inference time and model parameters are further elaborated in the time complexity section of the official comment. We will expand our discussion on complexity, considering both theoretical and practical system perspectives. Additionally, we will include details regarding the inference time and memory usage in the manuscript.

---

> > ### Comment · Reviewer_um1j · 2023-08-14
> > **Thank you for additional details**
> >
> > Thank you for providing additional details in the rebuttal. I think that most of my questions and concerns have been addressed.

---

> > > ### Author Response · Authors · 2023-08-15
> > > **Thanks for your response**
> > >
> > > We are glad to know that our response is satisfactory to you. We plan to include all the new experimental results in the manuscript as soon as we are permitted a polish for the final version.

---

### Official Review · Reviewer_YmDt · 2023-07-05

**Soundness:** 2 fair
**Presentation:** 2 fair
**Contribution:** 2 fair
**Rating:** 4
**Confidence:** 5

**Summary:**

In this work, the authors propose to incorporate features from five-body interaction into machine-learning force field models and develop QuinNet. To efficiently incorporate such high-order information, the authors are motivated by the topology of many-body interactions and design sophisticated components. Experiments on several benchmarks are conducted to demonstrate the performance of QuinNet.

**Strengths:**

1. The target problem of this paper, the development of machine learning force field models, is of great significance.

**Weaknesses:**

1. **The motivation for the designed components of many-body interaction is puzzling**. As introduced in Section 4, the development of four-body interaction (improper torsions) and five-body interactions are based on the topology. First, such analysis is purely qualitative. The authors did not provide further completeness proof or quantitative evidence about these interaction schemes in real-world data. Second, the reasons for deriving Eq (4)-(9) are not well explained. It is suggested to clarify how these components are motivated according to the topology analysis.


2. **On the experimental evaluation**. Additionally, there are several aspects of the experiments that are concerned:
    - The empirical performance is not consistently better than other baselines. Among the evaluated benchmarks, the proposed QuinNet cannot outperform the baselines significantly. For example, in MD17, the newly developed five-body interaction modules do not significantly improve performance. In rMD17, the best performance is diversely distributed among the compared models. Overall, the experimental evaluation does not well demonstrate the power of newly developed modules.
    - The computation efficiency evaluation is missing. Although the authors provide complexity analysis, it is better to further show the time/memory cost comparison between the proposed QuinNet and baselines. Besides, the model parameters should also be provided for all compared models.
    - The scale of the chosen benchmarks is rather small. Both the dataset size and sample size (number of atoms) are limited. It is suggested to further evaluate the proposed QuinNet on large-scale benchmarks, e.g., Open Catalyst Project [1].
    - The ablation study. First, as shown in Figure 5, the inclusion of Five-body@I even induces further errors, which would make readers curious about whether such a phenomenon generally exists. Second, as introduced in VisNet, the improper angle was also considered. The authors should add further discussions and empirical comparisons between it and the newly proposed four-body interaction (improper torsion).


3. **The writing does not meet the requirement of an acceptable paper in this conference**. First, Section 3.2 can be thoroughly extended (e.g., in the appendix) to introduce the background of force fields and highlight the importance of torsion potential, improper torsions, and higher-order many-body interactions. Second, there lack of formal descriptions of QuinNet. Figure 3 can hardly be understood by readers that are not familiar with the related works in this area.

[1] Chanussot L, Das A, Goyal S, et al. Open catalyst 2020 (OC20) dataset and community challenges[J]. Acs Catalysis, 2021, 11(10): 6059-6072.
    -

**Questions:**

Please refer to the Weakness section to address the concerns.

**Limitations:**

The authors did not discuss the limitations of this work.

---

> ### Author Rebuttal · Authors · 2023-08-10
>
> We thank the reviewer for his/her comments and will address each point in our response accordingly.
>
> ### Weakness 1:
> * In our experiments, we employ Chignolin, a protein system [1,2], which offers quantitative evidence regarding the impact of five-body interactions, aligning with the conclusions in Ref [3]. As stated in Ref [4], "it is unclear how many descriptor elements are actually needed in order to make the descriptor complete and thus able to uniquely specify an atomic environment of the N neighbors." Moreover, from the perspective of "Many-Body Expansion" theory [5], representing the energy of an entire system in a hierarchical level requires including all many-body interactions for completeness.
>
> [1] van der Spoel, David, and M. Marvin Seibert. "Protein folding kinetics and thermodynamics from atomistic simulations." Physical review letters 96.23 (2006): 238102.
>
> [2] Satoh, Daisuke, et al. "Folding free‐energy landscape of a 10‐residue mini‐protein, chignolin." FEBS letters 580.14 (2006): 3422-3426.
>
> [3] Wang, Jiang, et al. "Multi-body effects in a coarse-grained protein force field." The Journal of Chemical Physics 154.16 (2021).
>
> [4] Bartók A P, Kondor R, Csányi G. On representing chemical environments[J]. Physical Review B, 2013, 87(18): 184115.
>
> [5] Collins M A, Bettens R P A. Energy-based molecular fragmentation methods[J]. Chemical reviews, 2015, 115(12): 5607-5642.
>
> * Thanks for your suggestion. As illustrated in Figure 2 of the manuscript, three-body interactions can be represented as angles (Fig. 2a). To efficiently calculate the physical quantity, we can adopt the method used in PaiNN. Similarly, the approach employed in ViSNet can be utilized to compute dihedral angles by calculating the normal vectors of planes $ijk_1$ and $ijk_2$, and the inner product of these two normal vectors will yield the torsion angle (Fig. 2b). For improper angles, we can first compute the normal vector of the plane $ij_1j_2$, then find the inner product of $ij_3$ and the normal vector (Fig. 2c). Five-body interactions follow a similar process, involving the calculation of normal vectors for planes $ij_1j_2$ and $ij_3j_4$ in Fig. 2d; planes $ij_1k_1$ and $ij_2k_2$ in Fig. 2e; and planes $ik_1k_2$ and $ijk_3$ in Fig. 2f. The inner product of these two normal vectors will then provide the dihedral angle. We will include more details in the manuscript to better explain our methodology.
>
> ### Weakness 2:
> * MD17 and revised MD17 are small molecular datasets in which the influence of five-body interactions is relatively minor. Nevertheless, five-body interactions have been demonstrated to be crucial in various scenarios, such as replicating specific phenomena in protein systems. Consequently, QuinNet is not only compatible with other state-of-the-art models on MD17 and rDM17 benchmarks without any sacrifices but also outperforms numerous leading models on larger molecular systems, such as the MD22 and Chignolin datasets.
> * Time Complexity section in the official comment showcases the complexity of explicitly calculating the relevant physical quantities, as well as the calculations performed in QuinNet. Additionally, the inference time and model parameters are presented in the Time Complexity section of the official comment.
> * As highlighted in the introduction section, to demonstrate the significance of five-body interactions, it is essential to test large-scale systems. The mean system size in OC20 is 77.75 (as reported in ComENet), whereas the size of supramolecules in MD22 ranges from 42 to 370, and the size of the Chignolin protein is 166. Experimental results indicate that five-body interactions have a more substantial impact on larger molecules than on smaller ones, such as those in the MD17 and rMD17 datasets. Furthermore, in response to other reviewers' suggestions, we have included the performance results for the QM9 dataset in Table 3 of the official comment section to demonstrate our model's effectiveness with varying dataset sizes. Due to time constraints, the experiments are not yet fully converged. Therefore, the table presents the current results, and we will update it once the experiments are completed. We will also incorporate these results into the manuscript.
> * First, we arrange the 5-body interactions as depicted in Figure 2(d)-(e) since 5-body interactions@I encompass a portion of 4-body (improper) interactions and 5-body interactions@III include a part of 6-body interactions. According to our ablation study, when incorporating the 5-body@I component, the energy errors increase, whereas the force errors remain comparable to the results of 4-body (improper) interactions. Although 5-body interactions@I can partially represent the improper term with dihedral angles, using dihedral angles for this purpose may not be suitable, as the descriptions for the improper term in empirical force fields shown in Figure 1 typically involve height or angles. These inappropriate pieces of information may potentially damage performance to some extent. Second, it should be noted that ViSNet incorporates the improper angle in its latest version on arXiv, which was released after the NeurIPS submission deadline and has not yet undergone peer review. Consequently, the related discussion was not included.
>
> ### Weakness 3:
> * We will thoroughly revise the paper to improve its clarity and readability. The revisions will include:
>   * Providing a more detailed introduction to the background, which will help readers who are not familiar with the field to gain a better understanding of the concepts.
>   * Offering a more comprehensive context description for Figure 3, ensuring that the information presented is clear and easily interpretable.

---

### Official Review · Reviewer_8RW7 · 2023-07-06

**Soundness:** 3 good
**Presentation:** 3 good
**Contribution:** 2 fair
**Rating:** 5
**Confidence:** 5

**Summary:**

This paper aims to incorporate 5-body interactions into geometric deep learning models. They first analyze the topology of 5-body interactions and identify three 5-body angles. Then they propose an efficient way to incorporate these 5-body information into models. The complexity of the proposed QuinNet is still O(|N|), the same as many previous 2-body methods like PaiNN. The results are comparable to previous SOTA methods.

**Strengths:**

This paper is well-written and easy to follow.

The experimental results show that the proposed method can perform well on most tasks. The ablation study in Section 5.4 and Figure 5 show that the proposed 5-body information indeed helps to model.

**Weaknesses:**

See details in the Question part.

**Questions:**

1. About the motivation:
 this paper aims to incorporate 5-body interactions into geometric deep learning models. However, based on my understanding, using up to 4-body (torsions) interaction is already complete [1][2] in terms of capturing the geometric structures. If this is correct, then why do we need these 5-body angles? In addition, if we can incorporate 5-body interactions, do we also need to incorporate 6-body interactions?

2. About the complexity:
in Section 4.3, the authors claim that the complexity is O(|N|), as efficient as many 2-body methods like SchNet and PaiNN. But I think this complexity is not well explained. Using pseudocode/algorithm may be better to analyze the complexity. In addition to the analysis, I suggest the authors use some results to empirically verify the great efficiency compared to other baseline methods, e.g. the inference time, used memory, etc.

3. About the tasks:
this paper focuses on MLFFs, how about other molecular property prediction tasks, such as QM9 and OC20? I am wondering if this method is specially designed for MLFFs, or can be used on all 3D molecule tasks. In other words, why do the authors emphasize MLFFs? Is there any significant difference between MLFFs and other molecule property prediction tasks?

4. Other related papers: many-body [3], MLFFs [4]

5. The j, k in Figure 2 are confusing to me. For example, in (f), why not be i, j1, j2, j3, and k1?

[1] ComENet: Towards Complete and Efficient Message Passing for 3D Molecular Graphs.
[2] GemNet: Universal Directional Graph Neural Networks for Molecules.
[3] On the Expressive Power of Geometric Graph Neural Networks.
[4] Forces are not Enough: Benchmark and Critical Evaluation for Machine Learning Force Fields with Molecular Simulations.

---

> ### Author Rebuttal · Authors · 2023-08-10
>
> We thank the reviewer for his/her comments and will address each point in our response accordingly.
>
> ### Questions 1:
> * **4-body interactions are not complete for modeling molecular interactions.** As stated in Ref [1], "it is unclear how many descriptor elements are actually needed in order to make the descriptor complete and thus able to uniquely specify an atomic environment of the N neighbors." Moreover, from the perspective of "Many-Body Expansion" theory [2], representing the energy of an entire system in a hierarchical level requires including all many-body interactions for completeness. ComENet demonstrates its geometric completeness for a strongly connected 3D graph, proving that torsion angles are indeed adequate for capturing geometric structures. However, determining the identical nature of two 3D graphs does not guarantee the ability to model molecular interactions accurately. Furthermore, while incorporating 5-body interactions may be over-complete for capturing geometric structures, our results indicate their effectiveness with only a minor increase in complexity. The inclusion of 6-body interactions could potentially lead to increased computational costs if deemed necessary. Nevertheless, due to the computational complexity and the challenge of identifying an appropriate physical quantity to describe many-body interactions, this area warrants further investigation. We hope our work can offer valuable insights for future studies.
>
> [1] Bartók A P, Kondor R, Csányi G. On representing chemical environments[J]. Physical Review B, 2013, 87(18): 184115.
>
> [2] Collins M A, Bettens R P A. Energy-based molecular fragmentation methods[J]. Chemical reviews, 2015, 115(12): 5607-5642.
>
> ### Questions 2:
> * Time Complexity section presented in the official comment part, illustrates the complexity of calculating relevant physical quantities explicitly and within the QuinNet framework. Additionally, the inference time and model parameters are detailed in the Time Complexity section of the official comment part.
>
> ### Questions 3:
> * Machine learning force fields are indeed an important scientific problem. Our network has the potential to naturally extend to molecular property prediction tasks. In response to your suggestion, we have conducted experiments on the QM9 dataset, and the preliminary results are displayed in Table 3 of the official comment. Due to time constraints, the experiments have not yet converged. Therefore, the table presents the current results, and we will update it once the experiments are completed. We will also incorporate these results into the manuscript. In comparison to molecular property prediction tasks, which typically predict a single value, machine learning force fields demand a more significant trade-off between accuracy and efficiency. This is the rationale behind designing our network with a complexity of $O(|N|)$.
>
> ### Questions 4:
> * We will add them into references.
>
> ### Questions 5:
> * $j$ and $k$ represent the neighbor atoms of $i$ and $j$, respectively. We will revise the figures and symbols to enhance the clarity of the manuscript.

---

### Official Review · Reviewer_YYfR · 2023-07-07

**Soundness:** 3 good
**Presentation:** 3 good
**Contribution:** 3 good
**Rating:** 7
**Confidence:** 4

**Summary:**

This paper introduces a machine learning force field that is a neural network with explicit interactions for up to 5-body terms.  The authors evaluate the model on a couple of public datasets and show demonstrate the competence or superiority of this new model compared to the state of the art in this field.

**Strengths:**

The paper provides an important addition to a series of ever-improving machine learning potentials.  The contribution is clear and simple to understand at the high level, though the details are often unclear.  The benchmarks were compared against a set of reasonably strong published methods in this area.  In my opinion, if this work was presented in an unambiguously clear fashion and accompanied by code, it could be a strong contribution to this conference.

[The paper improved significantly following the first round of feedback from reviewers, so I'm raising my rating to a 7.]

**Weaknesses:**

The complexity analysis is very limited.  How many total interactions did the typical molecule have as a function of their atoms, and how did the practical experimental complexity scale for the evaluation of these molecules.  One of the main reasons that 5-body terms were not used in traditional MD simulations was the poor scaling of the number of interactions one would need to calculate.

The MD simulation mentioned in section 5.1 and Fig 4 are not described anywhere.  The following sentences suggest that there would be some explanations in the supplement, but I couldn't find them: "Additionally, we perform MD simulations using trained QuinNets as force fields and plot the distribution of interatomic distances h(r) for these 7 molecules in Fig. 4. Further details regarding additional settings can be found in the Supplementary Materials."

These sentences in the supplement, page2, are confusing or wrong: "Similarly, five-body@III interaction (Fig. S1 (c)) is a special case of six-body interaction when nodes i and k4 in Fig. S1 (d) superpose each other. Thus, the QuinNet model captures all five-body interactions and a portion of six-body interactions, making it a versatile and comprehensive tool for modeling complex molecular systems."  There is no six-body interaction if two of the bodies are the same, and there is no physically acceptable case where two different atoms could superpose each other.

The code is not provided, so it is not possible for me to assess the reproducibility of this method.  The diagram in Figure 3 seems reasonable at the very high level, but it lacks the definitions of most of the terms annotated in the figure, thus rendering it confusing.  (What is $Y_l$? is it the set of all spherical harmonics $Y_{lm}$ for a given angular momentum $l$? What is $n_j$? What is $s_j$?  $W$?...)

**Questions:**

Could the authors add the presentation of the QM9 quantities estimated in the recent publication for Allegro?  (https://www.nature.com/articles/s41467-023-36329-y Table 3)

How long and how stable were the actual MD simulations?  What were the exact codes/protocols used?

What is the practical performance of the model during evaluation?

**Limitations:**

No potential negative societal impacts from this work.

---

> ### Author Rebuttal · Authors · 2023-08-10
>
> We thank the reviewer for his/her comments and will address each point in our response accordingly.
>
> ### Weakness 1:
> * We address your concern through both theoretical and practical analyses. In the official comment part, we present the time complexity analysis and comparisons of inference time and model parameters. In empirical force fields, interactions are computed explicitly using physical quantities. For instance, two-body interactions are represented as bonds, with the order of the number of two-body interactions being $O(NN_b)$, where $N$ and $N_b$ denote the number of atoms and the number of neighbors, respectively. Three-body interactions are represented as angles, with an order of $O(NN_b^2)$. Four-body and five-body interactions are depicted as dihedral angles, with complexity orders of $O(NN_b^3)$ and $O(NN_b^4)$, respectively. Taking aspirin as an example, and setting the radius cutoff at 5Å, the number of two-body interactions, three-body interactions, four-body interactions (torsion), four-body interactions (improper torsion), five-body interactions@I, five-body interactions@II, and five-body interactions@III are 306, 2202, 38553, 10369, 35635, 628197, and 517155, respectively. **The number of five-body interactions is significantly greater than that of other many-body interactions.** This is why 5-body terms are typically not used in traditional MD simulations. However, in some cases, five-body interactions play a crucial role in various fields such as coarse-grained protein force fields, organic molecules, crystal vibrations, and electrostatic interaction potentials. The method introduced in QuinNet may provide an efficient solution in these cases.
>
> ### Weakness 2:
> * We apologize for the lack of detail in the Supplementary Materials. Simulations were carried out for each model and molecule, starting from the first frame configurations and spanning 300 ps. A 0.5 fs time step was employed, and the temperature was maintained at 500 K using a Nosé-Hoover thermostat. The distribution of interatomic distances, h(r), was computed as the ensemble average of distance statistics within the trajectories. The relevant descriptions will be defeinitely added to the Supplementary Materials.
>
> ### Weakness 3:
> * As shown in Equation 9, there are six indices, i.e., $i$, $j$, $j_1$, $j_2$, $k_1$, and $k_2$, where $j, j_1, j_2\in \mathcal{N}_i$ and $k_1, k_2 \in \mathcal{N}_j$. Therefore, Equation 9 describes six-body interactions. However, since $i \in \mathcal{N}_j$, $k_1$ or $k_2$ might be the same index as $i$, causing the equation to describe 5-body interactions, as there will be only five indices in the equation. This is why we state that the five-body@III interaction is a special case of six-body interaction. We will modify the sentence as "Similarly, five-body@III interaction (Fig. S1 (c)) is a special case of six-body interaction (Fig. S1 (d)) when the index i and k4 are the same in the Equation 9. Thus, the QuinNet model captures all five-body interactions and a portion of six-body interactions, making it a versatile and comprehensive tool for modeling complex molecular systems."
>
> ### Weakness 4:
> * The code will be released upon the paper's acceptance. To avoid confusion, we will add descriptions for the notations in the caption of Figure 3. $Y_l$ is indeed the set of $Y_{lm}$. $n_j$ represents the normal vector associated with node $j$, and $s_j$ denotes the scalar embedding of node $j$. $W$ refers to the weights in a linear layer, while $b$ indicates the bias.
>
> ### Questions 1:
> * We have conducted benchmarks on the QM9 dataset, and the preliminary results (Table 3) can be found in the official comment part. Due to time constraints, the experiments have not yet fully converged. Therefore, the table presents our current results, and we will update it once the experiments are completed. Additionally, we will include these results in the manuscript.
>
> ### Questions 2:
> * The MD simulations for molecules in the MD17 dataset were performed over 300 ps, and all seven trajectories exhibited stability. As shown in Refs. [1] and [2], simulations using most of the models remain stable for the MD17 dataset. Moreover, the code for the simulation was implemented using the ASE Python package and adapted from Ref. [1].
>
> [1] Fu X, Wu Z, Wang W, et al. Forces are not Enough: Benchmark and Critical Evaluation for Machine Learning Force Fields with Molecular Simulations[J]. Transactions on Machine Learning Research, 2023.
>
> [2] Wang Z, Wu H, Sun L, et al. Improving machine learning force fields for molecular dynamics simulations with fine-grained force metrics[J]. The Journal of Chemical Physics, 2023, 159(3).
>
> ### Questions 3:
> * In terms of practical performance, we conducted MD simulations for seven molecules in the MD17 dataset over 300 ps, and all trajectories exhibited full stability. Regarding accuracy, the distributions of interatomic distances h(r) calculated using the trained QuinNet model closely matched the results obtained from DFT calculations. As for efficiency, we have included additional analysis (Time complexity section) in the official comment part to demonstrate the performance of the QuinNet model.

---

> > ### Comment · Reviewer_YYfR · 2023-08-18
> > **Thanks for the response.**
> >
> > I have read the responses and appreciate the additional effort by the authors.  The response to weakness 3 is confusing: would one consider that a 2 body term captures a part of the three body terms or a 3-body term a part of the 4-body term? (these are all clearly orthogonal considerations and can vary independently; moreover there is never a case when two particles are on top of each other.)  I would urge the authors to put some additional effort into clarifying this sentence or dropping this point, however, I don’t feel strongly about it.

---

> > > ### Author Response · Authors · 2023-08-19
> > > **Thanks for your response**
> > >
> > > Thank you for your comments. In Weakness 3, our primary emphasis was on analysis from the formula's perspective, rather than from the actual system. As a result, there would be no unphysical situations involving overlapping atoms. Nevertheless, we acknowledge the reviewer's concerns, and recognize that the description in this section may create confusion for readers. Therefore, we opt to remove this point per your suggestion. We sincerely appreciate your valuable feedback, which significantly enhances the quality of our article.

---

### Author Response · Authors · 2023-08-10
**Reponse to all reviewers**

We are grateful for the valuable feedback provided by all the reviewers. Given that several concerns were commonly shared, we will address these questions collectively in this section.
### **Time Complexity**
The list below presents a comparison of the computational complexity for calculating physical quantities using both explicit methods and QuinNet. Here, $N$ and $N_b$ denote the number of atoms and the number of neighbors, respectively. Note that the number of atoms $N$ is ignored in the complexity analysis of QuinNet, .
* 3-body:
  * Complexity for calculating physical quantities: For each atom $i$, choose two neighbor atoms $j$ and $k$ to calculate angles. $NC_{N_b}^2\sim O(NN_b^2)$
  * Pseudocode in QuinNet:
$m_i = 0$
For $j \in\mathcal{N}_i$:

      &emsp;$m_i += r_{ij}$

    $h_i = m_i^2$
  * Complexity in QuinNet: $O(N_b)$
* 4-body (torsion):
  * Complexity for calculating physical quantities: For each atom $i$, choose two neighbor atoms $j$ and $k$ firstly, then choose one neighbor atom $l$ of atom $j$.$NC_{N_b}^2C_{N_b} \sim O(NN_b^3)$
  * Pseudocode in QuinNet:
$h_{ij,1} = 0, h_{ij,2} = 0$
For $k_1 \in \mathcal{N}_i$:

    &emsp;$h_{ij,1} += r_{ik_1} \times r_{ij}$

    For $k_2 \in \mathcal{N}_j$:

    &emsp;$h_{ij,2} += - r_{jk_2} \times r_{ij}$

    $h_{ij} = h_{ij,1}*h_{ij,2}$
  * Complexity in QuinNet: $O(N_b)$
* 4-body (improper):
  * Complexity for calculating physical quantities: For each atom $i$, choose three neighbor atoms $j$, $k$, and $l$. $NC_{N_b}^3 \sim O(NN_b^3)$
  * Pseudocode in QuinNet:
$m_i = 0, h_{i,1} = 0, h_{I,2} = 0$
For $j \in \mathcal{N}_i$:

    &emsp;$m_i += r_{ij}$
    &emsp;$h_{i,1} += \alpha_j * r_{ij}$
    &emsp;$h_{i,2} += \beta_j * r_{ij}$

    $h_i = m_i * (h_{i,1} * h_{i, 2})$
  * Complexity in QuinNet: $O(N_b)$
* 5-body@I:
  * Complexity for calculating physical quantities: For each atom $i$, choose four neighbor atoms $j$, $k$, $l$, and $m$. $NC_{N_b}^4\sim O(NN_b^4)$
  * Pseudocode in QuinNet:
$h_{i,1} = 0, h_{i,2} = 0$
For $j \in \mathcal{N}_i$:

    &emsp;$h_{i,1} += \alpha_j * r_{ij}$
    &emsp;$h_{i,2} += \beta_j * r_{ij}$

    $h_i = (h_{i,1} * h_{i, 2})^2$
  * Complexity in QuinNet: $O(N_b)$
* 5-body@II:
  * Complexity for calculating physical quantities: For each atom $i$, choose two neighbor atoms $j_1$ and $j_2$, then choose one neighbor atom $k$ for atom $j$. $NC_{N_b}^2C_{N_b}^1C_{N_b}^1\sim O(NN_b^4)$
  * Pseudocode in QuinNet:
$h_{j,1} = 0, h_{j,2} = 0, h_{i} = 0$
For $k \in \mathcal{N}_j$:

    &emsp;$h_{j,1} += \alpha_k * r_{kj}$
    &emsp;$h_{j,2} += \beta_k * r_{ij}$

    For $j \in \mathcal{N}_i$:

    &emsp;$h_{i} += (h_{j,1} \times h_{j,2})$

    $h_{i} = h_{i}^2$
  * Complexity in QuinNet: $O(N_b)$
* 5-body@III:
  * Complexity for calculating physical quantities: For each atom $i$, choose three neighbor atoms $j_1$, $j_2$ and $j_3$, then choose one neighbor atom $k$ for one of the three neighbor atoms of $i$. $NC_{N_b}^3C_{N_b}^1\sim O(NN_b^4)$
  * Pseudocode in QuinNet:
$h_{j,1} = 0, h_{j,2} = 0$
For $k \in \mathcal{N}_j$:

    &emsp;$h_{j,1} += \alpha_k * r_{kj}$
    &emsp;$h_{j,2} += \beta_k * r_{kj}$

    $h_{j} = h_{j,1} \times h_{j,2}$

    For $j \in \mathcal{N}_i$:

    &emsp;$h_{ij} = h_{i} * h_{j}$
  * Complexity in QuinNet: $O(N_b)$
Additionally, we evaluated the inference time of different models with a single 32GB V100 GPU card (Table 1). As the system size increases, the NequIP and Allgro models encounter out-of-memory issues. In general, the inference time of QuinNet is lower compared to the other benchmarked models.

Tabel1: (unit: ms)
|# atoms/model|100|150|200|250|300|350|400|
|--|--|--|--|--|--|--|--|
|NequIP|447±3.44|958±3.56|-|-|-|-|-|
|Allegro (1 layer)|94.9±0.48|200±1.07|347±4.12|534±1.02|767±1.19|-|-|
|Allegro (3 layers)|83.4±0.47|292±1.67|640±4.22|-|-|-|-|
|QuinNet|74.1±5.15|90.1±1.05|108±0.46|127±0.51|149±3.36|168±0.71|187±1.92|

The model parameters can be found in the table below:

Table 2:
|GemNet|NequIP|Allegro (1 layer)|Allegro (3 layers)|ViSNet|QuinNet|
|--|--|--|--|--|--|
|2.2M|3M|7M|18M|10M (estimate*)|9M|

\* The source code for ViSNet is currently unavailable. The model's parameter is estimated based on the hyperparameters described in the ViSNet paper.

### **Additional experiments on QM9 dataset**
We conducted benchmarks using the QM9 dataset, and the experimental results are presented below:

Table 3:
||units|Allegro|Equiformer|ViSNet|QuinNet|
|--|--|--|--|--|--|
|$\mu$|D|-|0.014|0.010|0.771|
|$\alpha$|$a_0^3$|-|0.056|0.041|0.047|
|Homo|meV|-|17|17.3|20.4|
|LUMO|meV|-|16|14.8|17.6|
|Gap|meV|-|33|31.7|28.2|
|$R^2$|$a_0^2$|-|0.227|0.030|0.194|
|ZPVE|meV|-|1.32|1.56|1.26|
|$U_0$|meV|4.7|10|4.23|7.6|
|$U$|meV|4.4|11|4.25|8.4|
|$H$|meV|4.4|10|4.52|7.8|
|$G$|meV|5.7|10|5.86|8.5|
|$C_v$|$\frac{cal}{mol K}$|-|0.025|0.023|0.024|

The table above displays the preliminary results due to time constraints. We will update the table once the experiments are fully converged.

---

### Decision · Program_Chairs · 2023-09-21

**Decision:**

Accept (poster)

**Comment:**

The paper received mixed ratings before the rebuttal.  Some common concerns shared by reviewers include inadequate complexity analysis, unclear motivation and some unclear model descriptions. The authors provide detailed complexity analysis, additional experimental results, and model clarifications. The authors’ response clarified some of the shared concerns and steered reviewer **8RW7** towards positive.

Though the paper could benefit from further clarification of the motivation and model description, the reviewers in general appreciate the paper’s effort in studying an important problem (including the reviewer **YmDt**, who expresses lukewarm negativity) and the convincing empirical model evaluations. The ACs tend to be on the positive side and agree that the paper could contribute valuable insights to the machine learning community, thus recommending acceptance. In the final version, wherever possible, the clarity should be further improved by incorporating well-articulated motivation, and model description.